# Changes in Mental Health among Japanese University Students during the COVID-19 Era: Differences by College Department, Graduate Level, Sex, and Academic Year

**DOI:** 10.3390/healthcare12090902

**Published:** 2024-04-26

**Authors:** Emma Underwood, Ryo Horita, Nanako Imamura, Taku Fukao, Miho Adachi, Satoko Tajirika, Ricardo Izurieta, Mayumi Yamamoto

**Affiliations:** 1College of Public Health, University of South Florida, Tampa, FL 33620, USA; 2Health Administration Center, Gifu University, Gifu 501-1193, Japanyamamoto.mayumi.t7@f.gifu-u.ac.jp (M.Y.); 3Gifu University Hospital, Gifu University, Gifu 501-1193, Japan; 4Medical Education Development Center, Gifu University, Gifu 501-1193, Japan; 5United Graduate School of Drug Discovery and Medical Information Sciences, Gifu University, Gifu 501-1193, Japan

**Keywords:** CCAPS-Japanese, mental health, undergraduate students, graduate students, sex differences

## Abstract

This study compared the effects of academic year, college department, and undergraduate or graduate status on Japanese students’ mental health during the COVID-19 pandemic. From 2021–2023, an online survey was conducted using the Counseling Center Assessment of Psychological Systems-Japanese (CCAPS-Japanese) to evaluate students’ mental health; 9395 undergraduate students (4623 female, 4772 male) and 1169 graduate students (380 female, 789 male) responded. Undergraduate students in medicine had lower levels of depression, generalized anxiety, and social anxiety than those in other departments. Engineering students exhibited the highest level of academic distress. First-year students had the highest levels of generalized and social anxiety but the lowest level of academic distress. Second-year students had the lowest level of depression, and third-year students had the highest level of academic distress. Among graduate students, first-year students had higher levels of depression, generalized anxiety, social anxiety, academic distress, and hostility than second-year students. Undergraduates had poorer mental health than graduate students. Females had higher levels of eating concerns than males among undergraduate students. This study revealed that the mental health of university students was affected by various factors. These findings demonstrate the characteristics of university students requiring early support.

## 1. Introduction

Globally, colleges are faced with a rise in mental health disorders and illnesses [1,2]. Most mental health disorders start before age 30. Early interventions, such as during college age, can reduce severity and prevent secondary disorders [3]. The World Health Organization (WHO) and the World Mental Health International College Student (WMH-ICS) Initiative revealed that the onset of mental disorders in college students occurs before college entry and remains untreated [4]. Early detection and treatment of mental health disorders in college is important, but only 25% of students seek help for emotional problems [5]. Barriers to mental health care for Japanese students include stigma, low mental health literacy, and attitudinal barriers. They exhibit a low perceived need for care and a desire to solve problems without outside help [6]. 

The WHO and WMH-ICS have demonstrated associations between stress and mental disorders in college students [7]. Graduate school students are vulnerable to academic stress that can affect their mental health due to high demands and competitive environments [8,9]. In one study, 39% of graduate students indicated having moderate to severe depression symptoms, a significant percentage higher than the general population, only 6% of whom scored within the same range [10]. Little is known about the prevalence of mental health disorders among graduate and undergraduate college students. There is also a lack of research on the impact of college department and major on students’ mental health status. College students’ mental health needs to be analyzed for better support. In addition, there are limited reports on how COVID-19 restrictions have affected mental health among higher education students in Japan [11] and Europe [12,13]. 

This study analyzed mental health survey data to compare numerous elements, such as grade, college department, and undergraduate or graduate status, and their effects on mental health among Japanese students amid the COVID-19 pandemic. We also wished to elucidate the features of students’ mental health during COVID-19.

## 2. Materials and Methods

### 2.1. Study Design

We analyzed mental health data collected over three years (2021–2023) through an online survey of students at Gifu University, Gifu, Japan. It was conducted as a mental health screening during the university’s annual health checkup, which is mandatory for all students. The university provided this based on Japan’s School Health and Safety Act. Students’ background information, including department, sex, and grade, was also collected from the annual health checkup data, following the STROBE guidelines [14].

### 2.2. Participants and Procedures

The participants included undergraduate and graduate students from Gifu University in 2021, 2022, and 2023. Gifu University is a national general university located in Gifu City, Japan. The school size, with approximately 7000 students, and historical background are average among the 800 higher education institutes in Japan. The area of Gifu City is approximately 10,621 square kilometers and it is located in the center of Japan. Written informed consent was obtained from all the participants at the start of the study. Students excluded from the study were doctoral graduate students, international students, those who had “other” as their department and year, those marked “other” for sex, those in their fifth or sixth year of undergraduate studies, and those over the age of 30 years. This was to increase the homogeneity of the research target. The School of Social System Management was also excluded because it is a new department that opened in 2021. The survey was conducted from February for all students and April for first-year students, as the academic year commences in April. After receiving the online survey via email, students were given 24 h to complete it on their personal computers or smartphone devices. Students were instructed to complete the survey by selecting numerical responses, as described below in Section 2.3—Measures.

### 2.3. Measures

This online survey used the Counseling Center Assessment of Psychological Systems (CCAPS-Japanese) to measure students’ psychological symptoms within the two weeks before the survey [15]. The CCAPS was initially developed in 2001 by the University of Michigan’s counseling and psychological services as the first multidimensional measure to assess psychological issues in college students [16]. The CCAPS-Japanese has been adapted and validated for Japanese students and has a rigorous factor structure, good internal consistency, adequate convergent validity, and good test–retest validity [15,17]. It contains 55 items with eight-factor derived subscales: depression (11 items), generalized anxiety (9 items), social anxiety (6 items), eating concerns (8 items), family distress (6 items), academic distress (3 items), hostility (7 items), and alcohol use (5 items). Responses were rated on a five-point Likert scale ranging from 0 (not at all like me) to 4 (extremely like me). Higher scores reflect higher levels of distress and psychological symptoms.

### 2.4. Data Analysis

A cross-sectional design compared CCAPS-Japanese with the eight subscale scores for various elements. One-way multivariate analysis of variance (MANOVA) was used to assess the relationship between the independent variables (college department, grade level as an undergraduate student, and academic year) and the dependent variables. A post-hoc Tukey analysis assessed the significance of the relationship between the independent variables and mental health status. The eta-squared value was used to measure effect size. A score of 0 to <0.01 was deemed “little to no effect”, 0.01 to <0.06 was a “small effect”, 0.06 to <0.14 was a “moderate effect”, and ≥0.14 was a “large effect”. For dichotomous variables such as sex and grade level (Master’s student and undergraduate versus graduate students), an independent samples *t*-test was performed to assess the relationship between the independent variable and mental health status. Cohen’s d was recorded as a measure of effect size, with 0.0 to <0.2 being “little to no effect”, 0.2 to <0.5 being a “small effect”, 0.5 to <0.8 being a “moderate effect” and ≥0.8 being a “large effect”. The results were considered statistically significant at *p* < 0.05 (two-sided). The covariates evaluated in this analysis were sex, age, university year, and undergraduate or graduate student status. The Graduate School of Natural Science and Technology offers a Master’s program in Applied Biological Sciences and Engineering. Therefore, considering homogeneity, we compared Applied Biological Sciences and Engineering undergraduate students with Graduate School of Natural Science and Technology graduates. Data were analyzed using Statistical Package for Social Sciences (SPSS) software version 28 (IBM, Armonk, NY, USA). 

### 2.5. Ethical Considerations

This study was approved by the Research Ethical Committee of the Graduate School of Medicine, Gifu University, Gifu, Japan (approval number: 2021-B114). Written informed consent was obtained from all the participants at the start of the study. They were notified that all their responses would remain confidential and anonymous. No compensation was provided to participants. The participants were informed that their participation would be unrelated to their academic evaluation and would not receive research participation credits.

## 3. Results

### 3.1. Participants’ Demographics

The participants’ characteristics according to grade and academic year are summarized in Table 1 and Table 2. The total number of students who responded to the survey in 2021–2023 was 11,215 (5300 female, 5915 male; mean = 20.45, SD = 3.399). There were no missing data. After applying the exclusion criteria, 10,564 students were included in this study (5003 female, 5561 male; mean = 20.00, SD = 1.785). Regarding participants, 9395 were enrolled in undergraduate programs (4623 female, 4772 male; mean = 19.62, SD = 1.470), while 1169 students were in a Master’s program (380 female, 789 male; mean = 23.05, SD = 0.993).

### 3.2. Department

The results of one-way MANOVA and post-hoc analysis by department are summarized in Table 3. The one-way MANOVA showed that the department in which a student was enrolled had a statistically significant relationship with all eight subscales among undergraduate students. However, the eta-squared statistic reflecting effect size was indicative of no effect in all subscales except for social anxiety (F(4, 9390) = 32.602, *p* ≤ 0.001, η^2^ = 0.014), academic distress (F(4, 9390) = 54.620, *p* ≤ 0.001, η^2^ = 0.023), and eating concerns (F(4, 9390) = 45.954, *p* ≤ 0.001, η^2^ = 0.019). Effect sizes were within small ranges. Post-hoc analysis showed that those in the Department of Medicine had lower levels of depression, generalized anxiety, and social anxiety than those in other departments. Engineering students exhibited the highest level of academic distress.

### 3.3. Grade

The results of one-way MANOVA and post-hoc analysis by grade for undergraduate students are summarized in Table 4. One-way MANOVA showed a statistically significant relationship between grade level and all eight subscales among undergraduate students. A small effect size was seen for academic distress (F(3, 9391) = 64.341, *p* ≤ 0.001, η^2^ = 0.020), and a moderate effect size was observed for alcohol use (F(3, 9391) = 358.639, *p* ≤ 0.001, η^2^ = 0.103). This may indicate age, as Japan’s legal drinking age is 20. Second-year students had the lowest depression levels, while first-year students had high anxiety levels. Third-year students had high academic distress levels.

The results of the independent samples *t*-test by grade for graduate students are summarized in Table 5. For Master’s students, an independent samples *t*-test showed a statistically significant relationship between grade level and all eight subscales except for alcohol use. The effect size for these subscales was small, except for eating concerns and family distress, for which no effect was observed.

### 3.4. Undergraduate versus Graduate

The results of independent samples *t*-tests comparing undergraduate and graduate students are summarized in Table 6. Applied Biological Sciences, Engineering, and Graduate School of Natural Science and Technology students at both levels were included in the analysis. An independent samples t-test showed a statistically significant relationship between undergraduate and graduate-level status in generalized anxiety, social anxiety, academic distress, eating concerns, and alcohol use. The effect size was small for the academic distress and alcohol use subscales.

### 3.5. Sex

The results of the independent samples *t*-test by sex for undergraduate students are summarized in Table 7. The independent samples *t*-test showed a statistically significant relationship between sex and seven out of eight of the subscales in undergraduate students: depression, generalized anxiety, social anxiety, academic distress, eating concerns, family distress, and alcohol use. Most subscales were in the no-effect range, but eating concerns were in the moderate-effect range. Females scored higher than males on all subscales except for alcohol use and academic distress.

The results of the independent samples *t*-test by sex for graduate students are summarized in Table 8. Among Master’s students, the independent samples *t*-test showed females scoring higher in depression, generalized anxiety, social anxiety, eating concerns, hostility, and family distress. Cohen’s d reflected a small effect size. Males scored higher than females in alcohol use. However, the effect size indicated no effect.

### 3.6. Academic Year

The results of one-way MANOVA and post-hoc analyses by academic year for undergraduate students are summarized in Table 9. For undergraduate students, the one-way MANOVA showed a statistically significant relationship between academic year and depression (F(2, 9392) = 16.722, *p* ≤ 0.001, η^2^ = 0.004), generalized anxiety (F(2, 9392) = 7.957, *p* ≤ 0.001, η^2^ = 0.002), social anxiety (F(2, 9392) = 13.017, *p* ≤ 0.001, η^2^ = 0.003), hostility (F(2, 9392) = 7.891, *p* ≤ 0.001, η^2^ = 0.002), and alcohol use (F(2, 9392) = 15.551, *p* ≤ 0.001, η^2^ = 0.003). Although significant at *p* ≤ 0.001, the effect size had little to no effect on all subscales.

The results of one-way MANOVA and post-hoc analysis by academic year for graduate students are summarized in Table 10. For Master’s students, the one-way MANOVA showed a statistically significant relationship between academic year and depression (F(2, 1166) = 6.389, *p* = 0.002, η^2^ = 0.011), generalized anxiety (F(2, 1166) = 6.092, *p* = 0.002, η^2^ = 0.010), social anxiety (F(2, 1166) = 7.105, *p* ≤ 0.001, η^2^ = 0.012), academic distress (F(2, 1166) = 5.881, *p* = 0.003, η^2^ = 0.010), and hostility (F(2, 1166) = 3.121, *p* = 0.044, η^2^ = 0.005). The effect size was small for all subscales except hostility, which had no effect. Post-hoc analysis showed that 2021 was the highest for all eight subscales compared to 2022 and 2023.

## 4. Discussion

This study aimed to examine mental health differences among Japanese university students based on factors such as department, grade level, sex, and academic year. The goal was to provide useful insights into students’ mental well-being and support the development of early intervention systems on campus.

Social anxiety and eating concerns are related to college enrollment among undergraduates. Medical students tend to have lower social anxiety, possibly due to having clear career goals and less fear of dropping out. Students may choose a career with fewer social components if they have social anxiety. Meanwhile, the Engineering department reported the lowest eating concerns. The Engineering department has mostly male students, while eating concerns were found to be higher among females in this study, which will be discussed later in this study.

Engineering students experience the highest academic distress compared to other departments due to the difficulty of the material and high demands. This aligns with studies showing that science-based degree programs, like engineering, receive more parental encouragement and emotional support [18,19].

The current research is limited to college department and major and their effects on the mental health of undergraduate students. However, the available literature focuses on depression, the only subscale in this study [20,21,22]. Studies have found that humanities majors rank higher in depression than science, technology, engineering, and mathematics (STEM) majors. Among STEM majors, engineering ranks higher in depression than medicine [20,21,22]. Our study found that a high severity of depression was observed in regional studies, engineering, and medicine. However, the effect size of college department on depression was found to be insignificant.

Third-year undergraduate students experience the highest level of academic distress compared to first- and second-year students. This suggests a progression in difficulty, as academic distress increases with each year. Academic distress decreased in graduate school after a point of adjustment to the curriculum in the fourth year of undergraduate education, which scored lower than the third year.

First-year Master’s students had higher levels of depression, generalized anxiety, social anxiety, academic distress, and hostility than second-year students. The influence of the survey period should be considered regarding these results. First-year Master’s students were surveyed in April, at the beginning of the academic year, which may reflect the stress of initial adaptation to the graduate school environment. Conversely, second-year Master’s students were surveyed in February, long before the submission of their Master’s theses. Therefore, their academic distress and accompanying depression and anxiety may have been relatively low.

Undergraduate students have poorer mental health compared to graduate students, except for alcohol consumption. Master’s students have a more structured academic and social life, with mentors and labs that provide a built-in support system. Positive relationships with mentors are linked to better mental well-being and use of mental health services [23]. Most undergraduate students lack mentor support and may feel isolated due to larger class sizes and lack of lab co-workers’ support.

Females have higher eating concerns than males, with a moderate effect size. This is supported by modern literature [24]. Social media promotes unrealistic beauty standards that affect young women’s body image and lead to dissatisfaction [25]. This is important to understand, as disordered eating in women positively correlates with suicidal ideation and suicidality [26].

Graduate-level students’ mental health status was influenced only by the academic year they took the online survey. The severity of depression, anxiety, and academic distress was higher in 2021 compared to 2022 or 2023, possibly due to the uncertainty and anxiety caused by the COVID-19 pandemic [27] and lack of a support system for students to deal with it. Undergraduates reported more stress and less positive mood, while older students reported less anxiety and depression than younger peers, conflicting with other research [28]. This may be because Master’s students would have experienced more interruptions to their academic careers because of the pandemic.

This study advances the understanding of university students’ mental health differences affected by department, grade, sex, and academic year. These findings are consistent with previous studies [29,30,31,32] demonstrating the characteristics of university students who need early support.

### Limitations and Future Directions

This study had some limitations. First, the generalizability of our study findings may be unclear, since our study collected data from only one university. Second, most effect sizes in this study were small, with the only moderate effect sizes being between grade and alcohol, and sex and eating concerns. This may indicate confounding factors that affected the mental health of university students. Third, the dataset used did not follow the same students throughout their academic careers. Instead, it introduced new cohorts in each academic year. Although this is a limitation, we believe the cohorts reasonably represent the sample population.

To improve this study, future analyses could include more variables such as familial history of mental illness, social support systems, financial status, sexual orientation, and whether the students or their loved ones had had COVID-19 recently. The stressors of adulthood combined with a rigorous college life can worsen a student’s mental health. Factors that correlate with mental illness include being female, having separated or deceased parents, having no religious affiliation, graduating from the bottom 70% of their high school class, living alone, having low economic status, and identifying as non-heterosexual [1,33,34]. Additional research must be conducted to better understand college students’ lifestyle choices and behaviors, mental and physical health, and how these factors interact.

## 5. Conclusions

This study adds to the existing literature by showing that mental health symptoms in college students can depend on various contributing extrinsic factors. Our findings show that depression, anxiety, academic distress, eating concerns, and other aspects that affect mental well-being are influenced by collegiate program-level status, enrollment department, grade level, and sex. We hope that our study will aid professionals in conceptualizing further research and public health programs to support students with their mental health needs along their collegiate journey.

## Figures and Tables

**Table 1 healthcare-12-00902-t001:** Participants’ background characteristics by grade.

	U1	U2	U3	U4	M1	M2	Total
Sex							
Male	1503 (53.2)	1075 (49.5)	995 (48.9)	1199 (50.8)	161 (57.9)	628 (70.5)	5561 (52.6)
Female	1324 (46.8)	1098 (50.5)	1038 (51.1)	1163 (49.2)	117 (42.1)	263 (29.5)	5003 (47.4)
Age							
Mean ± SD (Years)	18.26 ± 0.792	19.09 ± 0.899	20.15 ± 0.925	21.30 ± 0.917	22.59 ± 1.364	23.19 ± 0.792	20.00 ± 1.785
Department							
Applied Biological Sciences	507 (17.9)	416 (19.1)	365 (18.0)	397 (16.8)	0 (0.0)	0 (0.0)	1685 (16.0)
Education	540 (19.1)	468 (21.5)	477 (23.5)	462 (19.6)	25 (9.0)	74 (8.3)	2046 (19.4)
Engineering	1054 (37.3)	765 (35.2)	709 (34.9)	834 (35.3)	0 (0.0)	1 (0.10)	3364 (31.8)
Graduate School of Natural Science and Technology	0 (0.0)	0 (0.0)	0 (0.0)	0 (0.0)	243 (87.7)	798 (89.6)	1041 (9.9)
Medicine	488 (17.3)	334 (15.4))	301 (14.8)	454 (19.2)	2 (0.7)	3 (0.3)	1582 (15.0)
Regional Studies	238 (8.4)	190 (8.7)	181 (8.9)	215 (9.1)	7 (2.5)	15 (1.7)	846 (8.0)
Total	2827 (26.7)	2173 (20.6)	2033 (19.2)	2362 (22.4)	278 (2.6)	891 (8.4)	10,564

SD = standard deviation.

**Table 2 healthcare-12-00902-t002:** Participants’ demographics by academic year.

	2021	2022	2023	Total
Sex				
Male	1073 (48.0)	2135 (52.5)	2353 (55.2)	5561 (52.6)
Female	1161 (52.0)	1934 (47.5)	1908 (44.8)	5003 (47.4)
Age				
Mean ± SD (Years)	19.97 ± 1.972	20.08 ± 1.772	19.94 ± 1.747	20.00 ± 1.785
Year				
U1	736 (31.9)	934 (23.0)	1157 (27.2)	2827 (26.8)
U2	377 (16.9)	897 (22.0)	889 (21.1)	2173 (20.6)
U3	386 (17.3)	817 (20.1)	830 (19.5)	2033 (19.2)
U4	398 (17.8)	970 (23.8)	994 (23.3)	2362 (22.4)
M1	177 (7.9)	6.2 (1.5)	39 (0.9)	278 (2.6)
M2	160 (7.2)	389 (9.6)	342 (8.0)	891 (8.4)
Institute Level				
Undergraduate	1897 (84.9)	3618 (88.9)	3880 (91.1)	9935 (88.9)
Masters	337 (15.1)	451 (11.1)	381 (8.9)	1169 (11.1)
Department				
Applied Biological Sciences	370 (16.6)	638 (15.7)	667 (15.9)	1685 (16.0)
Education	448 (20.1)	827 (20.3)	771 (18.1)	2046 (19.4)
Engineering	670 (30.0)	1205 (29.6)	1489 (13.9)	3364 (31.8)
Graduate School of Natural Science and Technology	291 (13.0)	296 (9.7)	354 (8.3)	1041 (9.9)
Medicine	276 (12.4)	662 (16.3)	644 (15.1)	1582 (15.0)
Regional Studies	179 (8.0)	341 (8.4)	326 (7.7)	846 (8.0)
Total	2234 (21.1)	4069 (38.5)	4261 (40.3)	10,564

SD = standard deviation.

**Table 3 healthcare-12-00902-t003:** Sample survey of undergraduate students’ CCAPS scores by college department.

	Applied Biological Sciences (n = 1685)	Education (n = 1947)	Engineering (n = 3362)	Medicine (n = 1577)	Regional Studies (n = 824)	F	Partial Eta Squared	Post Hoc Tukey HSD
	Mean (SD)	Mean (SD)	Mean (SD)	Mean (SD)	Mean (SD)			
CCAPS-Japanese Subscales								
Depression	0.91 (0.78)	0.86 (0.78)	0.85 (0.72)	0.75 (0.71)	0.95 (0.72)	13.888 **	0.006	Applied Biological Sciences > Medicine; Education > Medicine; Regional Sciences > Education; Regional Sciences > Engineering; Applied Biological Sciences > Engineering; Engineering > Medicine
Generalized Anxiety	1.01 (0.73)	0.89 (0.76)	1.00 (0.71)	0.88 (0.71)	1.08 (0.70)	12.182 **	0.005	Applied Biological Sciences > Medicine; Education > Medicine; Engineering > Education; Engineering > Medicine; Regional Studies > Medicine; Regional Studies > Education; Regional Studies > Engineering
Social Anxiety	1.96 (0.91)	1.73 (0.94)	1.85 (0.90)	1.69 (0.94)	2.01 (0.91)	32.602 **	0.014	Applied Biological Science > Education; Applied Biological Science > Engineering; Applied Biological Sciences > Medicine; Engineering > Education; Engineering > Medicine; Regional Studies > Engineering; Regional studies > Medicine; Regional Studies > Education
Academic Distress	1.25 (0.84)	1.21 (0.83)	1.49 (0.88)	1.20 (0.83)	1.21 (0.78)	54.620 **	0.023	Engineering > Applied Biological Sciences; Engineering > Education; Engineering > Medicine; Engineering > Regional Science
Eating Concerns	1.19 (0.81)	1.22 (0.85)	0.97 (0.69)	1.14 (0.84)	1.19 (0.78)	45.954 **	0.019	Applied Biological Sciences > Engineering; Education > Engineering; Education > Medicine; Medicine > Engineering; Regional Studies > Engineering
Hostility	0.58 (0.66)	0.63 (0.71)	0.62 (0.67)	0.52 (0.63)	0.66 (0.69)	9.388 **	0.004	Regional Studies > Applied Biological Sciences; Education > Medicine; Engineering > Medicine; Regional Studies > Medicine
Family Distress	0.70 (0.66)	0.75 (0.72)	0.74 (0.65)	0.67 (0.67)	0.82 (0.71)	8.189 **	0.003	Regional Studies > Applied Biological Sciences; Education > Medicine; Engineering > Medicine; Regional Studies > Engineering; Regional Studies > Medicine
Alcohol Use	0.19 (0.48)	0.20 (0.48)	0.23 (0.51)	0.25 (0.58)	0.20 (0.48)	4.861 **	0.002	Engineering > Applied Biological Sciences; Medicine > Applied Biological Sciences; Medicine > Education

MANOVA test; ** *p* < 0.001; SD = Standard Deviation, CCAPS-Japanese = Counseling Center Assessment Psychological Symptoms-Japanese.

**Table 4 healthcare-12-00902-t004:** Sample survey of undergraduate students’ CCAPS scores and grade year.

	U1 (n = 2827)	U2 (n = 2173)	U3 (n = 2033)	U4 (n = 2362)	F	Partial Eta Squared	Post-Hoc Tukey’s HSD
	Mean (SD)	Mean (SD)	Mean (SD)	Mean (SD)			
CCAPS-Japanese Subscales							
Depression	0.88 (0.73)	0.79 (0.71)	0.86 (0.74)	0.87 (0.78)	7.378 **	0.002	U1 > U2; U2 < U3; U2 < U4
Generalized Anxiety	1.06 (0.72)	0.90 (0.70)	0.96 (0.73)	0.99 (0.74)	20.088 **	0.006	U1 > U2; U1 > U3; U1 > U4; U2 < U4
Social Anxiety	1.90 (0.92)	1.80 (0.92)	1.82 (0.92)	1.78 (0.93)	9.375 **	0.003	U1 > U2; U1 > U3; U1 > U4
Academic Distress	1.14 (0.76)	1.36 (0.85)	1.48 (0.91)	1.34 (0.89)	64.341 **	0.020	U1 < U2; U1 < U3; U1 < U4; U3 > U2; U3 > U4
Eating Concerns	1.06 (0.75)	1.15 (0.81)	1.14 (0.80)	1.10 (0.80)	7.209 **	0.002	U1 < U2; U1 < U3
Hostility	0.60 (0.67)	0.56 (0.65)	0.63 (0.68)	0.61 (0.69)	4.918 **	0.002	U3 > U2; U4 > U2
Family Distress	0.72 (0.69)	0.70 (0.66)	0.75 (0.68)	0.75 (0.68)	2.787 *	0.001	
Alcohol Use	0.04 (0.25)	0.10 (0.36)	0.38 (0.63)	0.40 (0.62)	358.639 **	0.103	U1 < U2; U1 < U3; U1 < U4; U2 < U3; U2 < U4;

MANOVA Test; * *p* < 0.05, ** *p* < 0.001; SD = Standard Deviation, CCAPS-Japanese = Counseling Center Assessment Psychological Symptoms-Japanese.

**Table 5 healthcare-12-00902-t005:** Sample survey of graduate students’ CCAPS scores and grade year.

	M1 (n = 278)	M2 (n = 891)	*t*	df	*p*-Value	Cohen’s *d*
	Mean (SD)	Mean (SD)				
CCAPS-Japanese Subscales						
Depression	0.99 (0.78)	0.76 (0.72)	4.512	1167	<0.001	0.310
Generalized Anxiety	1.18 (0.81)	0.99 (0.73)	3.474	429.009	<0.001	0.251
Social Anxiety	1.96 (0.93)	1.68 (0.91)	4.347	1167	<0.001	0.299
Academic Distress	1.25 (0.77)	1.08 (0.76)	3.275	1167	<0.001	0.255
Eating Concerns	1.08 (0.77)	1.00 (0.70)	1.584	430.682	0.057	0.114
Hostility	0.71 (0.70)	0.54 (0.62)	3.595	424.624	<0.001	0.262
Family Distress	0.78 (0.65)	0.69 (0.66)	2.049	1167	0.020	0.141
Alcohol Use	0.47 (0.68)	0.44 (0.65)	0.597	1167	0.275	0.041

*t*-test; SD = Standard Deviation, CCAPS-Japanese = Counseling Center Assessment Psychological Symptoms-Japanese.

**Table 6 healthcare-12-00902-t006:** Sample survey of CCAPS scores comparing undergraduate and graduate students.

	Undergraduate (n = 9395)	Graduate (n = 1169)	*t*	df	*p*-Value	Cohen’s *d*
	Mean (SD)	Mean (SD)				
CCAPS-Japanese Subscales						
Depression	0.80 (0.74)	0.82 (0.74)	1.450	10,562	0.073	0.045
Generalized Anxiety	0.98 (0.72)	1.03 (0.75)	−2.236	10,562	0.013	-0.069
Social Anxiety	1.83 (0.92)	1.75 (0.92)	2.815	10,562	0.002	0.087
Academic Distress	1.31 (0.86)	1.12 (0.76)	8.007	1556.137	<0.001	0.227
Eating Concerns	1.11 (0.79)	1.02 (0.72)	4.001	1541.553	<0.001	0.115
Hostility	0.60 (0.67)	0.58 (0.64)	1.082	10,562	0.140	0.034
Family Distress	0.73 (0.68)	0.71 (0.65)	0.841	1498.425	0.200	0.025
Alcohol Use	0.22 (0.51)	0.45 (0.65)	−11.551	1350.272	<0.001	-0.434

**Table 7 healthcare-12-00902-t007:** Sample survey of undergraduate students’ CCAPS scores and sex.

	Male (n = 4772)	Female (n = 4623)	*t*	df	*p*-Value	Cohen’s *d*
	Mean (SD)	Mean (SD)				
CCAPS-Japanese Subscales						
Depression	0.80 (0.72)	0.91 (0.76)	−7.438	9321.733	<0.001	−0.154
Generalized Anxiety	0.97 (0.72)	1.00 (0.73)	−2.062	9393	0.020	−0.043
Social Anxiety	1.78 (0.93)	1.88 (0.92)	−5.195	9393	<0.001	−0.107
Academic Distress	1.35 (0.88)	1.28 (0.83)	3.500	9388.738	<0.001	0.072
Eating Concerns	0.89 (0.65)	1.34 (0.85)	−28.681	8609.235	<0.001	−0.594
Hostility	0.60 (0.68)	0.61 (0.67)	−0.706	9393	0.240	−0.015
Family Distress	0.71 (0.64)	0.75 (0.71)	−2.930	9225.949	0.002	−0.061
Alcohol Use	0.26 (0.55)	0.18 (0.46)	7.127	9162.453	<0.001	0.147

*t*-test; SD = Standard Deviation, CCAPS-Japanese = Counseling Center Assessment Psychological Symptoms-Japanese.

**Table 8 healthcare-12-00902-t008:** Sample survey of Master’s students’ CCAPS scores and sex.

	Male (n = 789)	Female (n = 380)	*t*	df	*p*-Value	Cohen’s *d*
	Mean (SD)	Mean (SD)				
CCAPS-Japanese Subscales						
Depression	0.74 (0.69)	0.97 (0.81)	−4.720	656.484	<0.001	−0.311
Generalized Anxiety	0.99 (0.72)	1.13 (0.81)	−2.766	677.276	0.003	−0.180
Social Anxiety	1.71 (0.91)	1.83 (0.93)	−2.099	1167	0.018	−0.131
Academic Distress	1.13 (0.76)	1.11 (0.78)	0.331	1167	0.370	0.021
Eating Concerns	0.93 (0.63)	1.20 (0.84)	−5.455	595.188	<0.001	−0.375
Hostility	0.53 (0.62)	0.68 (0.68)	−3.664	1167	<0.001	−0.229
Family Distress	0.65 (0.59)	0.85 (0.75)	−4.592	613.703	<0.001	−0.311
Alcohol Use	0.48 (0.67)	0.37 (0.61)	2.831	816.658	0.002	0.171

*t*-test; SD = Standard Deviation, CCAPS-Japanese = Counseling Center Assessment Psychological Symptoms-Japanese.

**Table 9 healthcare-12-00902-t009:** Sample survey of undergraduate students’ CCAPS scores by academic year.

	2021 (n = 1897)	2022 (n = 3618)	2023 (n = 3880)	F	Partial Eta Squared	Post-Hoc Tukey’s HSD
	Mean (SD)	Mean (SD)	Mean (SD)			
CCAPS-Japanese Subscales						
Depression	0.94 (0.75)	0.83 (0.73)	0.83 (0.74)	16.722 **	0.004	2021 > 2022; 2021 > 2023
Generalized Anxiety	1.05 (0.72)	0.96 (0.72)	0.98 (0.73)	7.957 **	0.002	2021 > 2022; 2021 > 2023
Social Anxiety	1.93 (0.90)	1.80 (0.93)	1.81 (0.93)	13.017 **	0.003	2021 > 2022; 2021 > 2023
Academic Distress	1.33 (0.83)	1.31 (0.85)	1.31 (0.86)	0.575	0.000	
Eating Concerns	1.09 (0.77)	1.10 (0.79)	1.13 (0.79)	1.884	0.000	
Hostility	0.65 (0.69)	0.57 (0.65)	0.60 (0.69)	7.891 **	0.002	2021 > 2022; 2021 > 2023
Family Distress	0.75 (0.68)	0.73 (0.67)	0.72 (0.69)	1.730	0.000	
Alcohol Use	0.17 (0.44)	0.21 (0.50)	0.25 (0.55)	15.551 **	0.003	2022 > 2021; 2023 > 2021; 2023 > 2022

MANOVA Test; ** *p* < 0.01; SD = Standard Deviation, CCAPS-Japanese = Counseling Center Assessment Psychological Symptoms-Japanese.

**Table 10 healthcare-12-00902-t010:** Sample survey of Master’s students’ CCAPS scores by academic year.

	2021 (n = 337)	2022 (n = 451)	2023 (n = 381)	F	Partial Eta Squared	Post-Hoc Tukey’s HSD
	Mean (SD)	Mean (SD)	Mean (SD)			
CCAPS-Japanese Subscales						
Depression	0.94 (0.75)	0.79 (0.75)	0.82 (0.74)	6.389 *	0.011	2021 > 2022; 2021 > 2023
Generalized Anxiety	1.15 (0.81)	1.01 (0.76)	0.96 (0.68)	6.092 *	0.010	2021 > 2022; 2021 > 2023
Social Anxiety	1.91 (0.93)	1.67 (0.91)	1.71 (0.92)	7.105 **	0.012	2021 > 2022; 2021 > 2023
Academic Distress	1.24 (0.79)	1.08 (0.75)	1.07 (0.75)	5.881 *	0.010	2021 > 2022; 2021 > 2023
Eating Concerns	1.05 (0.78)	1.02 (0.68)	0.99 (0.70)	0.534	0.001	
Hostility	0.65 (0.70)	0.55 (0.62)	0.54 (0.62)	3.121 *	0.005	
Family Distress	0.75 (0.69)	0.69 (0.62)	0.71 (0.66)	1.025	0.002	
Alcohol Use	0.45 (0.68)	0.46 (0.64)	0.43 (0.65)	0.249	0.000	

MANOVA Test; * *p* < 0.05, ** *p* < 0.01; SD = Standard Deviation, CCAPS-Japanese = Counseling Center Assessment Psychological Symptoms-Japanese.

## Data Availability

Data are contained within the article.

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
