# Peer review of "Changes in Mental Health among Japanese University Students during the COVID-19 Era: Differences by College Department, Graduate Level, Sex, and Academic Year"

_healthcare, 2024, doi:10.3390/healthcare12090902_

Round 1

Reviewer 1 Report

Comments and Suggestions for Authors

The authors conducted a cross-sectional study to evaluate college students’ mental health during covid-19. Overall the manuscript is well-written and this is an interesting study. I have a few comments that I think could help strengthen the presentation of the methods and results.

  • The generalizability of your study findings is unclear since all of your study participants were from Gifu University. Could you provide more information on the location of Gifu University? How representative are the students at Gifu University? Do they have distinct demographics compared to other universities in Japan?
  • lines 83-89 should be moved to the result section
  • In section 2.4, please indicate if the p-value is two-sided or one-sided.
  • In section 3.3, could you also compare the demographics between undergraduate and graduate? Other than age, are there any other demographics that are significantly different between undergraduate and graduate, such as sex? The differences in some demographic variables could explain the differences you observed between undergraduate and graduate.
  • Similarly, in section 3.5, could you compare the demographics between different academic years? Other than age, are any other demographics significantly different between academic years?
  • It is unclear if there were any participants included in your study have missing data. Did the 9,395 undergrad and the 1,169 graduate students all provide complete questionnaire data? If not, how did you handle the missing data? Please provide relevant information to clarify these.

Author Response

We deeply appreciate your supportive comments. We have edited our previous manuscript, and the responses to your suggestions are as follows. We have also highlighted the corresponding text in the manuscript in yellow font.

The generalizability of your study findings is unclear since all of your study participants were from Gifu University. Could you provide more information on the location of Gifu University? How representative are the students at Gifu University? Do they have distinct demographics compared to other universities in Japan?

Thank you for your comment. We have now added a description regarding the characteristics of Gifu University(section 2.2, lines 75-78). In addition, we have added information to the Limitations and Future Directions section regarding the generalizability of the results, as this is an area that could be improved in future studies(section 4.1, lines 315-316).

lines 83-89 should be moved to the result section

Thank you for your suggestion. We have now moved those sentences, as well as Tables 1 and 2, to the Results section(section 3.1, lines 138-151).

In section 2.4, please indicate if the p-value is two-sided or one-sided.

Thank you for your suggestion. We have now indicated that the p-value is two-sided(section 2.4, line 119).

In section 3.3, could you also compare the demographics between undergraduate and graduate? Other than age, are there any other demographics that are significantly different between undergraduate and graduate, such as sex? The differences in some demographic variables could explain the differences you observed between undergraduate and graduate.

All demographics were not statistically significant between undergraduate and graduate students, apart from age. Probably, because almost graduate students are coming from undergraduate of same university, Gifu University.

Similarly, in section 3.5, could you compare the demographics between different academic years? Other than age, are any other demographics significantly different between academic years?

All demographics were not significantly different between academic years. Age was also not significantly different. This is because each academic year included students across all grades.

It is unclear if there were any participants included in your study have missing data. Did the 9,395 undergrad and the 1,169 graduate students all provide complete questionnaire data? If not, how did you handle the missing data? Please provide relevant information to clarify these.

Thank you for your kind advice. We have now added that there were no missing data other than the exclusion criteria(section 3.1, lines 138-151).

Reviewer 2 Report

Comments and Suggestions for Authors

The text concerns an important issue, namely the psychological health of students. The severity of depression, social anxiety, generalized anxiety, academic stress, alcohol use, eating disorders, family distress, hostility was examined (dependent variables).

Many independent sociodemographic variables were included (year of study and level of study (graduate or undergraduate), field of study (medical, technical, humanities), gender and year of measurement of the variables (2021, 2022 and 2023).

The research, as the authors emphasize, was of a screening nature. The specificity of the study was the moment of its implementation, namely it covered the period of the Covid-19 pandemic, i.e. 2021 (the "acute phase" of the pandemic), 2022 and 2023 (the end of the pandemic).

The results confirmed differences (some of them were already known from other studies) in the intensity of indicators of students' psychological health (in terms of depression, anxiety, etc.) related to gender, field of study, stage of study and year of study, and indicated general differences related to the study period ( 2021, 2022 and 2023).

An interesting result indicates that some dependent variables (including depression etc.) were more severe in 2021 than in 2022 and than in 2023. Is this the effect of adaptation to the rigors of the Covid-19 pandemic, the effect of the support for students (if any) ?), or the effect of improving the health situation/controlling the pandemic? A more complete explanation would make it possible to provide data from earlier years - before the Covid-19 pandemic. The authors claim that the supervisors' closer care for master's students has a protective effect, to which they attribute the role of psychological support, or perhaps only in the educational/academic dimension (help in preparing the master's thesis) (?).

The authors explain the lower level of depression among students in their final year of studies due to the fact that they are finalizing their master's thesis. However, isn't finishing/finalizing your master's thesis stressful? Unless the students "see" the end of their studies, which increases their sense of well-being and improves their mood? Maybe it's cultural specificity.

Similar differences in the intensity of psychological health indicators - as in graduate studies - were observed in undergraduate studies - with the exception of drinking alcohol, which - in the latter cases - (generally speaking) increased from year to year. The fact that alcohol consumption was at a higher level in 2022 than in 2021 can possibly be explained by the inability to cope with the stress resulting from the threat to health and life caused by the pandemic, but how to explain the even higher level of alcohol consumption in 2023? Is it the specificity of the culture of student life at the undergraduate level (also in this case it would be useful to provide data from earlier years (if the authors have them)), or maybe it is a manifestation of "relief" in connection with the control of the pandemic?

The rigors for students due to the pandemic have not been clearly defined; Was there remote learning in 2021, 2022 and 2023? Was learning suspended temporarily? Were students constantly staying on the university premises in 2021-2022-2023? Providing such information could be interesting for readers from other cultures. Remote learning or breaks in learning may have resulted in increased anxiety about educational success for some students, and may have increased the sense of health security for others due to isolation from a large group.

I wonder whether a more synthetic approach to presenting research results is possible. It is worth considering the possibility of resigning from some data, the multiplicity of which makes it difficult to generalize the results.

Undoubtedly, the authors' conclusion about the need to expand the support system for students is justified. It is also justified to promote educational activities to change lifestyles, because we know little about the long-term effects of the Covid-19 pandemic.

Author Response

We deeply appreciate your supportive comments. We have edited our previous manuscript, and the responses to your suggestions are as follows. We have also highlighted the corresponding text in the manuscript in yellow font.

An interesting result indicates that some dependent variables (including depression etc.) were more severe in 2021 than in 2022 and than in 2023. Is this the effect of adaptation to the rigors of the Covid-19 pandemic, the effect of the support for students (if any) ?), or the effect of improving the health situation/controlling the pandemic? A more complete explanation would make it possible to provide data from earlier years - before the Covid-19 pandemic. The authors claim that the supervisors' closer care for master's students has a protective effect, to which they attribute the role of psychological support, or perhaps only in the educational/academic dimension (help in preparing the master's thesis) (?).

Thank you for your suggestion. We have now added to the Discussion following your suggestion(section 4, lines 304-305; section 4.1, lines 324).

The authors explain the lower level of depression among students in their final year of studies due to the fact that they are finalizing their master's thesis. However, isn't finishing/finalizing your master's thesis stressful? Unless the students "see" the end of their studies, which increases their sense of well-being and improves their mood? Maybe it's cultural specificity.

Thank you for your comment. Finishing a master's thesis is definitely stressful. However, this survey was conducted long before the master's thesis deadline. Therefore, we considered that the level of depression was low in second-year Master’s students.

Similar differences in the intensity of psychological health indicators - as in graduate studies - were observed in undergraduate studies - with the exception of drinking alcohol, which - in the latter cases - (generally speaking) increased from year to year. The fact that alcohol consumption was at a higher level in 2022 than in 2021 can possibly be explained by the inability to cope with the stress resulting from the threat to health and life caused by the pandemic, but how to explain the even higher level of alcohol consumption in 2023? Is it the specificity of the culture of student life at the undergraduate level (also in this case it would be useful to provide data from earlier years (if the authors have them)), or maybe it is a manifestation of "relief" in connection with the control of the pandemic?

Thank you for your insights. At the end of the pandemic, increased drinking opportunities may have resulted in a higher level of the alcohol use subscale. However, there is no evidence of this hypothesis from the present data, so it is not mentioned in the Discussion. Basically, the percentage of routine alcohol drinking in Japanese university students is less than 10%. It is very difficult to estimate the COVID-19 effect on drinking behavior.

The rigors for students due to the pandemic have not been clearly defined; Was there remote learning in 2021, 2022 and 2023? Was learning suspended temporarily? Were students constantly staying on the university premises in 2021-2022-2023? Providing such information could be interesting for readers from other cultures. Remote learning or breaks in learning may have resulted in increased anxiety about educational success for some students, and may have increased the sense of health security for others due to isolation from a large group.

Thank you for your comment. There may have been some remote learning, but its percentage was small in 2021, 2022, and 2023 in Gifu University. In addition, the percentage of remote learning varied by department and class. Therefore, it is difficult to consider the impact of remote learning.

I wonder whether a more synthetic approach to presenting research results is possible. It is worth considering the possibility of resigning from some data, the multiplicity of which makes it difficult to generalize the results.

Thank you for your comment. We have now added information to the Limitations and Future Directions section regarding the generalizability of the results(section 4.1, lines 315-316).

Reviewer 3 Report

Comments and Suggestions for Authors

Really interesting article!  I enjoyed reviewing it.  I do have some feedback for improvement, though:

- Why did you focus exclusively on Gifu University?  There's nothing wrong with selecting just one university, but given your article's discussion of Japan as a while, the article would benefit from an explanation as to why you only focused on one university.  (This can be 1-2 sentences; it doesn't need to be a lengthy explanation.)

- Was there any data collected on whether the students or their loved ones had had COVID recently?  I saw that CCAPS was developed in 2001, so it wouldn't have included questions on COVID, but did the Counseling Center ask students about this?  These data might be analyzed alongside CCAPS (I'm aware of the limitations of correlations vs. causations, these data might be interested starting points for future work.)

- I did see that CCAPS collected data on 'eating concerns'.  Did that category include eating disorders and/or food insecurity?  There's a lot of research on the relationships between food insecurity and mental health outcomes, and depending on the mental health classifications in Japan, eating disorders might be considered mental illnesses (they're considered as such in the United States). 

- You mentioned that first-year Master’s students had higher levels of depression, generalized anxiety, social anxiety, academic distress, and hostility than second-year students.  Did you collect any data on how far students were from their families and what their social support systems were?  Adapting to a new environment, especially one as competitive as a graduate program, can be incredibly challenging, so I was wondering how many students had social support to mitigate the negative mental health outcomes.

Author Response

We deeply appreciate your supportive comments. We have edited our previous manuscript, and the responses to your suggestions are as follows. We have also highlighted the corresponding text in the manuscript in yellow font.

- Why did you focus exclusively on Gifu University?  There's nothing wrong with selecting just one university, but given your article's discussion of Japan as a while, the article would benefit from an explanation as to why you only focused on one university.  (This can be 1-2 sentences; it doesn't need to be a lengthy explanation.)

Thank you for your comment. We have now added a description regarding the characteristics of Gifu University(section 2.2, lines 75-78). In addition, we have now added information to the Limitations and Future Directions section regarding the generalizability of the results(section 4.1, lines 315-316).

- Was there any data collected on whether the students or their loved ones had had COVID recently?  I saw that CCAPS was developed in 2001, so it wouldn't have included questions on COVID, but did the Counseling Center ask students about this?  These data might be analyzed alongside CCAPS (I'm aware of the limitations of correlations vs. causations, these data might be interested starting points for future work.)

Thank you for your helpful comment. We did not collect data on whether the students or their loved ones had had COVID-19 recently. We have added this point to the Limitations and Future Directions section(section 4.1, lines 324-325).

- I did see that CCAPS collected data on 'eating concerns'.  Did that category include eating disorders and/or food insecurity?  There's a lot of research on the relationships between food insecurity and mental health outcomes, and depending on the mental health classifications in Japan, eating disorders might be considered mental illnesses (they're considered as such in the United States).

Thank you for your comment. The Eating Concerns subscale does not include items related to food insecurity. The CCAPS asks about symptoms related to eating disorders, but it is not a diagnostic tool. Eating disorders are considered mental illnesses in Japan.

- You mentioned that first-year Master’s students had higher levels of depression, generalized anxiety, social anxiety, academic distress, and hostility than second-year students.  Did you collect any data on how far students were from their families and what their social support systems were?  Adapting to a new environment, especially one as competitive as a graduate program, can be incredibly challenging, so I was wondering how many students had social support to mitigate the negative mental health outcomes.

Thank you for your suggestion. We did not collect data on how far students were from their families and what their social support systems were. We have now added this point to the Limitations and Future Directions section(section 4.1, lines 324)